# Tabular Foundation Models Are Effectively Shallow

**Irene Cannistraci** [1]   **Julia E. Vogt** [1]

## Abstract

In-context-learning Tabular Foundation Models (TFMs) are uniquely suited to scenarios with limited resources since they require no additional training on the target data. However, their architectures impose high inference latency and memory costs, hindering deployment in some environments. Simpler alternatives (e.g. Gradient-Boosted Decision Trees) run on CPU and often match their performance, but at the cost of manual feature engineering, preprocessing, and hyperparameter tuning that transformer models avoid by design. In this paper we show that it is possible to simplify TFMs, by substituting up to ~94% of the blocks using a closed-form linear translator, while preserving downstream performance and requiring minimal compute. We evaluate our approach across 15 classification and regression benchmarks spanning both general and clinical domains. This provides a practical pathway to dramatic throughput enhancements and lightweight inference, preserving the zero-shot convenience of foundation models without the prohibitive computational costs.

## 1. Introduction

Tabular foundation models (e.g., TabPFNv2 (Hollmann et al., 2025), TabICL (QU et al., 2025), TabDPT (Ma et al., 2026), Mitra (Zhang et al., 2026)) are attractive especially in scenarios like clinical use: the training set is passed as context at inference time, so a single pre-trained model serves every downstream task without gradient updates on the target dataset. The inference footprint, however, blocks their adoption: 12–16 transformer blocks, with diverse pre-training and attention mechanisms, demand GPUs that on-premise clinical environments typically do not have. Simpler alternatives such as Light Gradient-Boosting Machine (LightGBM) and Gradient-Boosted Decision Trees (GB-DTs) run on CPU and often match or surpass deep tabular models on performance (Grinsztajn et al., 2022; McElfresh et al., 2023), but at a different cost: manual feature engineering, preprocessing, and per-dataset hyperparameter tuning that transformer models avoid by design. This similar performance suggests that TFM models could be over-parametrized, and that the redundancy observed in vision and language transformers (Dalvi et al., 2020; Cannistraci et al., 2024; Jacobs et al., 2026) might be present in this scenario as well. Tabular transformers, however, differ fundamentally from those settings: short sequences (8–74 tokens), heterogeneous feature types, making it non-obvious whether the same redundancy holds. In this paper we investigate whether TFMs are vastly overparametrized.

We evaluate four TFMs across 15 diverse datasets, covering the Grinsztajn benchmark (Grinsztajn et al., 2022), TabZilla controls (McElfresh et al., 2023), clinical screening and ICU mortality (Johnson et al., 2016; Pollard et al., 2018), and regression tasks across heterogeneous domains. We first establish the existence of this redundancy by measuring per-block sensitivity and pairwise representation similarity. Next, we empirically demonstrate that this redundancy can be exploited: replacing contiguous blocks with a closed-form linear translator (Cannistraci et al., 2024) fit on just 500 calibration samples recovers the downstream Area Under the Receiver Operating Characteristic Curve (AUROC). Remarkably, this approach allows for extreme compression; as shown in Table 1, we can drop all but a single transformer block (e.g., approximating 15 of 16 layers) and still match or even exceed the full model's performance, whereas naive layer dropping results in catastrophic failure.

## 2. Method

**Linear approximation error.** Let $\mathbf{H}_l \in \mathbb{R}^{N \times d}$ denote the representation at depth $l \in \{0, 1, \ldots, L\}$: $\mathbf{H}_0$ is the tokenizer output and $\mathbf{H}_l$ for $l \geq 1$ is the output of block $l$ (equivalently, the input to block $l+1$); $L$ is the model's number of blocks. Following Cannistraci et al. (2024), given a start depth $s$ and an end depth $e$ with $0 \leq s < e \leq L$, we ask how well the contiguous stack of blocks $s+1, \ldots, e$ (taking $\mathbf{H}_s$ as input and producing $\mathbf{H}_e$ as out-

---

[1]Department of Computer Science, ETH Zurich. Correspondence to: Irene Cannistraci <irene.cannistraci@inf.ethz.ch>.

Accepted at the 2nd ICML Workshop on Foundation Models for Structured Data (FMSD), Seoul, South Korea. Copyright 2026 by the author(s).

put) can be approximated by a single linear map. We define $\varepsilon(s, e)$ as the relative residual of the best-fit linear map from $\mathbf{H}_s$ to $\mathbf{H}_e$:

$$
\varepsilon(s, e) = \frac{\|\mathbf{H}_e - (\mathbf{H}_s\mathbf{W}^* + \mathbf{b}^*)\|_F}{\|\mathbf{H}_e\|_F},
$$
$$
(\mathbf{W}^*, \mathbf{b}^*) = \arg\min_{\mathbf{W}, \mathbf{b}} \|\mathbf{H}_e - (\mathbf{H}_s\mathbf{W} + \mathbf{b})\|_F^2, \quad (1)
$$

where $\mathbf{b}^*$ is broadcast across the $N$ rows. $(\mathbf{W}^*, \mathbf{b}^*)$ is computed in closed form via least squares on calibration representations extracted by forward hooks. A small $\varepsilon(s, e)$ means a single affine map closely reproduces what blocks $s+1, \ldots, e$ collectively compute, *i.e.*, those blocks are linearly redundant.

**Per-architecture translators.** Equation (1) fits a single $(\mathbf{W}, \mathbf{b})$ that is shared across all sequence tokens, which suits single-axis self-attention (TabICL, TabDPT). Dual-axis attention models (TabPFNv2, Mitra) attend along both items and features, so a single shared map mixes the two axes and destroys the per-feature correspondence between $\mathbf{H}_s$ and $\mathbf{H}_e$. For these models we instead fit a separate $(\mathbf{W}_f, \mathbf{b}_f)$ per feature index $f$ on tensors of shape $(\cdot, n_{\text{features}}, d)$, applied as $\mathbf{H}'[\cdot, f, \cdot] = \mathbf{H}[\cdot, f, \cdot]\mathbf{W}_f + \mathbf{b}_f$; for Mitra, whose blocks process the in-context support and query examples as a tuple of two such tensors, we additionally fit one per-feature translator per stream (Section A.5). With $n_{\text{features}}=1$ the per-feature parameterisation reduces exactly to the single-$(\mathbf{W}, \mathbf{b})$ case.

**Block approximation.** At inference time the trained affine translator $(\mathbf{W}^*, \mathbf{b}^*)$ replaces blocks $s+1, \ldots, e$: input $\mathbf{H}_s$ is mapped directly to $\mathbf{H}_e$, and the remaining blocks $e+1, \ldots, L$ together with the pre-trained classification head run unchanged. The least-squares solve take 33–109 ms on a single GPU (calibration size $N=500$ items) and require no gradients, backpropagation, or parameter updates. This operation has two uses: with $w = e - s = 1$ we replace a single block at a time to measure each block's individual importance (per-block sensitivity); with $w \geq 2$ we replace a contiguous stack of blocks at once for higher compression.

## 3. Experiments

**Setup.** We study four pre-trained TFMs with frozen weights: TabPFNv2, TabICL, TabDPT, and Mitra. We evaluate on 15 diverse datasets. Twelve are classification tasks from three sources: five from the Grinsztajn medium-tabular benchmark (Grinsztajn et al., 2022) (MagicTelescope, house_16H, pol, COMPAS, road-safety), three TabZilla controls (McElfresh et al., 2023) (Jannis, Higgs, Covertype, subsampled to 30K), and four clinical datasets (Cardiotocography, Thyroid, MIMIC-III, eICU-CRD). We

additionally evaluate three regression tasks (Grinsztajn house_sales and superconduct, plus MIMIC-III length-of-stay). Each dataset is split into train, calibration and test, where the calibration split is used to fit the linear translator and to select the skip range, never to evaluate. We report AUROC (macro-averaged one-vs-rest for multi-class) and $R^2$ for regression. Dataset details and GBDT hyper-parameters, as well as other implementation details are in Section A.

### 3.1. Analysis

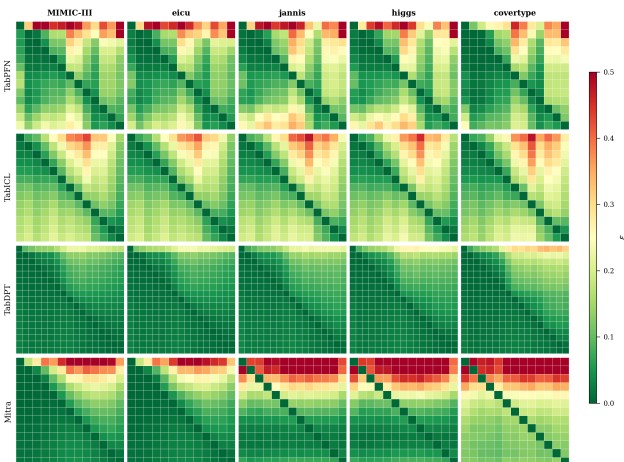

*Figure 1.* Pairwise linear approximation error $\varepsilon(s, e)$ (Equation (1)) on the largest clinical (MIMIC-III, eICU-CRD) and TabZilla (Jannis, Higgs, Covertype) datasets, and the four TFMs.

Figure 1 reports $\varepsilon(s, e)$ for all block pairs, where rows are the TFMs and columns the datasets. The figure shows that for some models (e.g., TabDPT) representations are more similar to each other, while for Mitra they are more difficult to approximate. In general, results are model dependent rather than dataset dependent, and the earlier blocks do most of the work. After the first blocks, TabDPT saturates completely with an $\varepsilon < 0.06$.

We then evaluate whether this behaviour reflects on the downstream AUROC. Therefore we select the skip window $(s, e)$ using the AUROC on the held-out calibration split: 500 items, the same validation split already used for hyper-parameter and model selection. The procedure consumes no additional patient data and never touches test labels.

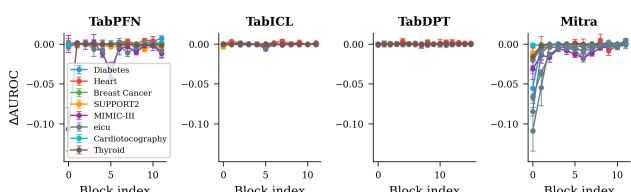

*Figure 2.* Per-block sensitivity ($\Delta$AUROC when one block is replaced by a linear translator). Uniform $y$-axis across panels.

Figure 2 replaces each individual block with a linear translator and measures ΔAUROC. TabICL and TabDPT are entirely flat ($|\Delta| < 0.005$): no individual block is critical. TabPFNv2 shows scattered sensitivity, while Mitra shows higher sensitivity in the first blocks.

Figure 3 shows the best approximation window distribution for each model when considering 3 or 4 blocks. Block 0 is preserved across nearly every dataset (yellow color), and TabDPT shows no clear preference for any approximation position over blocks 1 to $L-1$, indicating that for this model it is possible to pick *any* contiguous later window without per-dataset tuning.

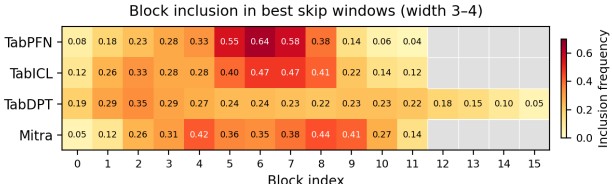

*Figure 3.* Block inclusion frequency in the best approximation windows (width 3-4) across datasets.

**Attention**    Figure 4 reports TabICL per-block mean attention entropy on surviving blocks before and after replacing a contiguous block window with a linear translator; the curves overlap closely highlighting that our approximation procedure does not meaningfully change the attention distribution. Meanwhile in Section B.5 we show that NLP transformer heads' attention distributions tend to differentiate across layers (Dalvi et al., 2020), whereas we show that tabular foundation model heads look comparatively flat across depth.

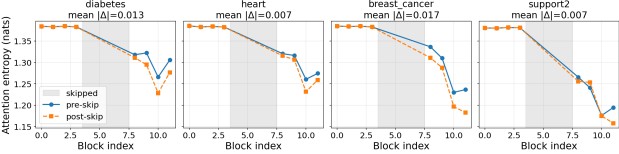

*Figure 4.* TabICL per-block mean attention entropy on surviving blocks before vs. after replacing blocks from 4 to 8 with a fitted linear translator (skip window shaded). Entropy in nats.

### 3.2. Results

Table 1 reports all regimes side-by-side. We tested approximating 3 or 4 blocks, 8 blocks and maximum-compression (retaining only a single block). Additionally we tested whether a simple identity function is enough instead of the linear approximator.

**Approximating 3 or 4 contiguous blocks (Best Approx.) preserves accuracy in most cases.** The best window is selected with the calibration-set AUROC as shown in Figure 3. On the numerical Grinsztajn datasets (pol, MagicTelescope, house_16H) and the multi-class clinical datasets (CTG, Thyroid) the approximation is essentially lossless: mild compression matches the full model to within a single test-AUROC point (e.g., TabICL on house_16H .958→.957). TFMs outperform GBDTs on Jannis (TabICL .866 vs. XGBoost .860) and remain competitive on TabZilla controls. TabICL outperforms GBDTs in almost all the settings, except for medical domains (i.e., MIMIC and eICU), where results are still comparable (i.e., .805 vs. .800 on MIMIC).

**Approx. max scales aggressively to a single retained block.** Remarkably, the models retain highly competitive accuracy even when discarding over 90% of their depth. The degradation follows a clear complexity gradient: the numerical Grinsztajn datasets and multi-class clinical tasks remain nearly lossless even under the most extreme compression (e.g., dropping 11 of 12 blocks on TabICL for the pol dataset drops AUROC only from .999 to .998; TabDPT dropping 15 of 16 blocks on CTG maintains .996 exactly). Meanwhile, MIMIC, eICU, COMPAS, road-safety, and the larger TabZilla benchmarks degrade gently at extreme widths. For TabDPT, the single retained block is block 0 on $11/12$ datasets, confirming that block 0 carries the dominant computation under single-axis self-attention; for TabICL it is block 0 on $11/12$. Under the per-feature translator that respects dual-axis attention, TabPFNv2 and Mitra select later blocks much more often (block 0 retained on $1/12$ and $3/12$ respectively): the translator equalises the contribution of different blocks, so any contiguous window can be replaced.

**The linear approximation is essential.** Naive layer dropping (*i.e.*, Identity) at the same maximal compression collapses the AUROC of every model. For example, dropping 11 of 12 blocks with Identity on TabPFN for the pol dataset plummets the AUROC to .419, whereas our closed-form linear translator (Approx. max) preserves it at .917. The contrast is similarly stark for Mitra on Covertype (Identity .301 vs. Linear .837) and on COMPAS (.431 vs. .671).

**Ablations.**    We additionally test whether compressed models remain robust under random feature corruption and structured clinical missingness (all-vitals or all-labs dropped) (Sections C.2 and C.3), and if per-patient $\varepsilon$ is uniform across outcome subgroups (Section C.1), confirming that compression does not introduce subpopulation-specific degradation. Furthermore, a natural alternative to block approximation is early exit: discarding the pre-trained head and refitting a logistic regression on intermediate representations. However, in Section C.4, we show that across all configurations, our block approximation, which retains the pre-trained head and bridges $\mathbf{H}_l \to \mathbf{H}_L$ with a linear translator, matches or beats early exit. Finally, we verify

*Table 1.* Downstream AUROC for GBDT baselines and four foundation models. Per model: **Full** = pre-trained model unchanged; **Best Approx.** = best contiguous approximation window of width 3–4 (mild compression); **Linear** ($w/L$) = linear translator at intermediate compression; **Approx. max** = linear translator at maximum compression (11 of 12 blocks approximated, or 15 of 16 for TabDPT); **Identity** = naive layer dropping at the same maximum compression (no translator). All approximation windows are selected by calibration-set AUROC. **Bold**: best per column. Underline: approximate variant $\geq$ Full. *Italic*: degrades $> 1\%$ from Full. First five datasets are from the Grinsztajn medium-tabular benchmark; next three are TabZilla controls; last four are clinical.

| Model | | MagicTel. | House16H | Pol | COMPAS | RoadSafe | Jannis[†] | Higgs | Covertype[‡] | CTG[†] | Thyroid[†] | MIMIC | eICU |
|---|---|---|---|---|---|---|---|---|---|---|---|---|---|
| XGBoost | | .934±.004 | .950±.003 | **.999**±**.000** | **.816**±**.005** | **.852**±**.002** | .860±.005 | .788±.007 | **.979**±**.001** | .997±.002 | **1.00**±**.000** | **.805**±**.007** | **.792**±**.005** |
| LightGBM | | .932±.006 | .950±.002 | **.999**±**.000** | .812±.008 | .849±.004 | .858±.004 | .789±.006 | .976±.002 | .998±.002 | **1.00**±**.000** | .804±.007 | .791±.005 |
| TabPFN | Full | .929 | .950 | **.999** | .777 | .843 | .846 | .780 | .957 | .998 | **1.00** | .758 | .758 |
| | Best Approx. | .928 | .949 | **.999** | .778 | .842 | .838 | .778 | .954 | .998 | .999 | .764 | .763 |
| | Linear (8/12) | .929 | .949 | **.999** | .777 | .838 | .842 | .778 | .952 | .997 | .999 | .764 | .760 |
| | Approx. max (11/12) | *.821* | *.841* | *.917* | .747 | *.743* | *.756* | *.621* | *.807* | *.983* | *.912* | *.710* | *.728* |
| | Identity (11/12) | *.479* | *.644* | *.419* | *.472* | *.531* | *.474* | *.463* | *.447* | *.460* | *.490* | *.441* | *.441* |
| TabICL | Full | **.939** | **.958** | **.999** | .804 | .845 | **.866** | **.790** | .962 | **.999** | **1.00** | .800 | .786 |
| | Best Approx. | .938 | .957 | **.999** | .798 | .836 | .863 | .787 | .961 | **.999** | **1.00** | .793 | .783 |
| | Linear (8/12) | .933 | .957 | **.999** | .782 | .826 | .858 | .785 | .959 | **.999** | .999 | .796 | .781 |
| | Approx. max (11/12) | .918 | .952 | .998 | .773 | .814 | .843 | .768 | .943 | **.999** | .999 | .785 | .777 |
| | Identity (11/12) | *.883* | *.912* | *.898* | *.713* | *.728* | *.697* | *.619* | *.659* | *.897* | *.853* | *.669* | *.695* |
| TabDPT | Full | .934 | .951 | .997 | .776 | .830 | .813 | .734 | .942 | .996 | .998 | .761 | .757 |
| | Best Approx. | .933 | .951 | .997 | .778 | .827 | .812 | .734 | .940 | .998 | .998 | .760 | .757 |
| | Linear (11/16) | .923 | .951 | .996 | .771 | .818 | .807 | .720 | .935 | .998 | .997 | .757 | .749 |
| | Approx. max (15/16) | *.905* | *.936* | *.985* | .765 | *.791* | *.792* | *.692* | *.918* | .996 | .990 | *.751* | .747 |
| | Identity (15/16) | *.532* | *.379* | *.568* | *.613* | *.546* | *.528* | *.487* | *.460* | *.578* | *.370* | *.527* | *.466* |
| Mitra | Full | .927 | .954 | .998 | .767 | .805 | .828 | .773 | .927 | .998 | .999 | .781 | .772 |
| | Best Approx. | .926 | .950 | .998 | .766 | .803 | .822 | .771 | .927 | .997 | .999 | .780 | .770 |
| | Linear (8/12) | .908 | .946 | .994 | .761 | .799 | *.801* | .741 | .922 | .996 | .999 | .770 | .759 |
| | Approx. max (11/12) | *.694* | *.867* | *.960* | *.671* | *.786* | *.694* | *.584* | *.837* | *.984* | *.945* | *.696* | *.708* |
| | Identity (11/12) | *.398* | *.439* | *.730* | *.431* | *.595* | *.509* | *.500* | *.301* | *.266* | *.555* | *.433* | *.409* |

[†] Macro-averaged OVR AUROC (CTG, Thyroid 3-class; Jannis 4-class). [‡] Covertype 7-class.

that block redundancy is not an artefact of classification by evaluating block approximation on three regression tasks (real-estate prices, materials science, ICU length-of-stay), where mild compression remains essentially lossless (Section B.6).

## 4. Conclusion and Future Work

In this work we showed that pre-trained TFMs (such as TabPFNv2, TabICL, TabDPT, and Mitra) are vastly over-parametrized for tabular tasks. By substituting up to ~94% of contiguous transformer blocks with a simple, closed-form linear translator (fitted on just 500 calibration samples) we successfully recovered downstream AUROC across diverse clinical and general tabular benchmarks. Remarkably, our approach allows state-of-the-art TFMs to be compressed to a single active transformer block, avoiding the catastrophic failure associated with naive layer dropping (Identity) and consistently outperforming early-exit strategies.

Our findings carry significant practical implications. By drastically reducing the computational footprint of TFMs, we bridge the gap between the zero-shot, tuning-free convenience of foundation models and the lightweight inference of traditional methods like GBDTs. This makes local, privacy-preserving deployment of modern tabular models more feasible for specific scenarios such as hospitals with

constrained hardware. Furthermore, our analysis of layer-wise sensitivity and attention entropy reveals fundamentally different representational dynamics compared to NLP or vision transformers, highlighting that early blocks perform the vast majority of the meaningful computation in tabular contexts.

**Future work.** The extreme compressibility of current TFMs naturally raises questions about their architectural design. Future work should investigate whether tabular foundation models can be explicitly pre-trained to be inherently shallower, or if this massive over-parametrization is a strict prerequisite for successful optimization during the pre-training phase. Additionally, we plan to extend our evaluation to real-time inference latency on standard commodity hardware, and explore whether structurally dynamic routing could allow TFMs to adapt their depth dynamically based on the complexity of individual data instances. Ultimately, unlocking lightweight, CPU-friendly inference for TFMs will be instrumental in bringing their zero-shot capabilities to widespread deployment in ubiquitous, resource-constrained environments.

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

# A. Implementation Details

## A.1. Datasets

**Grinsztajn medium-tabular benchmark** (Grinsztajn et al., 2022). MagicTelescope (binary, 19K / 10 numerical; gamma vs hadron). house_16H (22K / 16 numerical; house-price regression binarized by median threshold). pol (binary, 15K / 26 numerical). COMPAS-two-years (binary, 5K / 8 num + 9 cat; two-year recidivism). road-safety (binary, 13K / 29 num + 3 cat; road accident severity). Datasets loaded from OpenML (IDs 44125, 44123, 44122, 44162, 44161); subsampled to 30K when larger.

**Clinical (ICU mortality).** MIMIC-III (26,538 patients / 74 extracted features; 48-hour in-hospital mortality prediction) (Johnson et al., 2016). eICU-CRD (30,000 patients / 74 extracted features; multicenter ICU mortality prediction from 200+ U.S. hospitals) (Pollard et al., 2018).

**Clinical (multi-class).** Cardiotocography (CTG; 2,126 instances / 35 numerical features; 3-class fetal health classification). ann-thyroid (3,772 instances / 21 numerical features; 3-class thyroid function classification with substantial class imbalance).

**Non-medical TabZilla controls.** Jannis (4-class, 54 features), Higgs (binary, 28 features), and Covertype (7-class, 54 numerical/binary features), each subsampled to 30K instances for tractable in-context inference.

We use 68/12/20% train/validation/test splits. Numerical features are standardized using training-set statistics; categorical features are integer encoded; missing numerical values are imputed with zero after scaling. The validation split additionally serves as the block-approximation calibration set.

## A.2. GBDT Hyperparameters

XGBoost and LightGBM use standard defaults without per-dataset tuning: `n_estimators=500`, `max_depth=6`, `learning_rate=0.1`, `subsample=0.8`, `colsample_bytree=0.8`, with early stopping after 20 rounds on the validation set. These are intentionally not optimised: the goal is a reasonable GBDT baseline for context.

## A.3. Foundation Models

We use `n_estimators=1` for TabPFNv2 to enable deterministic per-block representation extraction (the ensembling of $n_{\text{est}}=8$ averages over independent forward passes, which would conflate per-block residuals across passes). Compressibility analysis measures *relative* block redundancy, which is invariant to the ensembling configuration. TabICL, TabDPT, and Mitra are used at their default configurations. All four models are pre-trained exclusively on synthetic data (TabPFNv2, TabICL) or non-overlapping real datasets (TabDPT's 123 real-data training sets, Mitra's curated real-data collection), ruling out data contamination as an explanation for the observed redundancy.

## A.4. Attention entropy

Where model architectures permit (TabICL), we measure attention diversity via per-head entropy: $\mathcal{H}(\mathbf{a}) = -\sum_j a_j \log a_j$, averaged across heads, queries, and samples. Entropy near $\log(S)$ (where $S$ is the sequence length) signals uniform attention; low entropy signals specialisation. TabPFNv2 and Mitra use dual-attention; TabDPT's hooks did not yield self-attention weights in our framework.

## A.5. Translator family per architecture

The *linear translator* of Cannistraci et al. (2024) is a closed-form affine map $\mathbf{H}_e \approx \mathbf{H}_s W + b$ that approximates the action of a contiguous stack of transformer blocks. The right parameterisation depends on the per-block representation structure of the underlying model. We instantiate it as follows for the four ICL foundation models in this paper:

- **TabICL, TabDPT (single-axis self-attention).** Block outputs are 2D, $(\text{seq}, d)$ (with optional pass/batch leading dimensions whose entries share the same block parameters). A single shared $(W \in \mathbb{R}^{d \times d}, b \in \mathbb{R}^d)$ fit by closed-form least squares on $(\text{seq} \cdots , d)$ flattened calibration data. $d^2 + d$ free parameters per skip.

- **TabPFNv2 (dual-axis attention).** Block outputs are $(n_{\text{items}}, n_{\text{features}}, d)$ from alternating between-items and between-features attention. Mixing the two axes destroys channel correspondence between source and target, so we fit a

separate ($W_f \in \mathbb{R}^{d \times d}, b_f \in \mathbb{R}^d$) per feature index $f$ and apply it as $\mathbf{H}'[\cdot, f, \cdot] = \mathbf{H}[\cdot, f, \cdot] W_f + b_f$. $n_{\text{features}} \cdot (d^2 + d)$ free parameters per skip.

- **Mitra (dual-axis 2D attention with separate streams).** Block outputs are a tuple ($\mathbf{H}^{\text{sup}}, \mathbf{H}^{\text{qry}}$), each of shape ($n_{\text{items}}, n_{\text{features}}, d$). The two streams share block parameters but have distinct activations (a translator that only updates support and bypasses query would put them at incompatible depths in subsequent blocks, producing spurious block-0 sensitivity not present in Mitra itself). We fit per-feature affines independently per stream: $2 \cdot n_{\text{features}} \cdot (d^2 + d)$ free parameters per skip.

With $n_{\text{features}} = 1$, the per-feature variant reduces exactly to the single-($W, b$) case, so this is a generalisation of the original linear translator rather than a different method. Concrete feature counts on our datasets: TabPFNv2's internal $n_{\text{features}}$ ranges from 10 (MagicTelescope) to 69 (Jannis); Mitra's covers a similar range across the Grinsztajn medium tier and clinical inputs.

**Conditioning.** Each per-feature solve has $n_{\text{items}}$ rows and $d{+}1$ columns. For TabICL/TabDPT the single ($W, b$) is fit on flattened ($n_{\text{items}} \cdot \text{seq}, d$), giving thousands of rows per skip — well over-determined. For TabPFNv2 ($d{=}192$, calibration capped at 500) the per-feature fit has $n_{\text{items}} {\geq} 500 \gg d{+}1{=}193$, also comfortably over-determined. For Mitra's support stream the fit uses the in-context training set (3,000 items per feature) and is over-determined; for Mitra's query stream ($d{=}512$, $n_{\text{items}} \leq 500$) it is marginally under-determined by at most 13 unknowns per output column. We use the `gelsd` driver, which returns the stable minimum-norm solution in either regime. The reported $\varepsilon$ is the translator's training residual on calibration activations and is therefore not a generalisation metric: generalisation is measured by test AUROC, and the translator never sees test data or any labels. All solves complete in $< 50$ ms per skip on a single GPU.

## B. Additional Results

### B.1. Why the saturation matrix is asymmetric

Each block writes an additive residual update $\Delta_{s \to e} = \mathbf{H}_e - \mathbf{H}_s$, so predicting $\mathbf{H}_s$ from $\mathbf{H}_e$ only requires subtracting a low-rank update (linearly easy), while predicting $\mathbf{H}_e$ from $\mathbf{H}_s$ requires synthesising that update from information not yet present (linearly hard whenever intermediate blocks injected non-linear content). The empirical asymmetry of Figure 1 is therefore a signature of the redundancy we exploit: later-block updates are predominantly low-rank corrections, not novel non-linear computation.

### B.2. Foundation model full results

Table 2 reports the per-(model, dataset, width) skip-compression result with the best skip start position selected by calibration-set AUROC. The corresponding aggregate "best width $\in \{3, 4\}$" numbers are in main-text Table 1. All models tolerate 3–4 approximated blocks with $|\Delta\text{AUROC}| \leq 2.0\%$ across the workshop dataset slice. The two ICU mortality datasets (MIMIC, eICU) and the mixed num+cat Grinsztajn tasks (COMPAS, road-safety) are the most resistant to extreme compression. At mild compression (width 3–4), position matters mostly via block 0: approximating block 0 is consistently the worst start; excluding start=0, TabICL and TabDPT have max test-AUROC spread of a few percentage points across (dataset, width). TabPFNv2 is more sensitive, making calibration-based selection essential.

*Table 2.* Foundation model block skipping: best skip position per (model, dataset, width). Linear translator. Selection by calibration-set AUROC.

| Model | Dataset | Width | Full AUROC | Skip AUROC | $\Delta$AUROC |
|---|---|---|---|---|---|
| | MagicTelescope | 3 | .929 | .928 | $-.001$ |
| | MagicTelescope | 4 | .929 | .918 | $-.011$ |
| | house_16H | 3 | .950 | .949 | $-.001$ |
| | house_16H | 4 | .950 | .949 | $-.001$ |
| | pol | 3 | .999 | .999 | $+.000$ |
| | pol | 4 | .999 | .999 | $-.000$ |
| | COMPAS | 3 | .777 | .777 | $-.000$ |
| | COMPAS | 4 | .777 | .778 | $+.001$ |
| | RoadSafety | 3 | .843 | .842 | $-.000$ |
| | RoadSafety | 4 | .843 | .842 | $-.001$ |
| | Jannis[†] | 3 | .846 | .838 | $-.008$ |
| TabPFN | Jannis[†] | 4 | .846 | .839 | $-.007$ |

*Table 2 continued*

| Model | Dataset | Width | Full AUROC | Skip AUROC | ΔAUROC |
|---|---|---|---|---|---|
| | Higgs | 3 | .780 | .778 | −.002 |
| | Higgs | 4 | .780 | .779 | −.001 |
| | Covertype$^{\ddagger}$ | 3 | .957 | .954 | −.003 |
| | Covertype$^{\ddagger}$ | 4 | .957 | .954 | −.002 |
| | CTG$^{\dagger}$ | 3 | .998 | .999 | +.001 |
| | CTG$^{\dagger}$ | 4 | .998 | .997 | −.000 |
| | Thyroid$^{\dagger}$ | 3 | 1.00 | .999 | −.000 |
| | Thyroid$^{\dagger}$ | 4 | 1.00 | .998 | −.001 |
| | MIMIC-III | 3 | .758 | .763 | +.005 |
| | MIMIC-III | 4 | .758 | .764 | +.006 |
| | eICU | 3 | .758 | .763 | +.004 |
| | eICU | 4 | .758 | .763 | +.004 |
| | MagicTelescope | 3 | .939 | .938 | −.001 |
| | MagicTelescope | 4 | .939 | .938 | −.001 |
| | house_16H | 3 | .958 | .957 | −.000 |
| | house_16H | 4 | .958 | .957 | −.000 |
| | pol | 3 | .999 | .999 | −.000 |
| | pol | 4 | .999 | .999 | −.000 |
| | COMPAS | 3 | .804 | .800 | −.004 |
| | COMPAS | 4 | .804 | .798 | −.006 |
| | RoadSafety | 3 | .845 | .836 | −.009 |
| | RoadSafety | 4 | .845 | .838 | −.007 |
| | Jannis$^{\dagger}$ | 3 | .866 | .863 | −.003 |
| TabICL | Jannis$^{\dagger}$ | 4 | .866 | .863 | −.003 |
| | Higgs | 3 | .790 | .787 | −.003 |
| | Higgs | 4 | .790 | .788 | −.002 |
| | Covertype$^{\ddagger}$ | 3 | .962 | .961 | −.001 |
| | Covertype$^{\ddagger}$ | 4 | .962 | .962 | −.001 |
| | CTG$^{\dagger}$ | 3 | .999 | .999 | +.000 |
| | CTG$^{\dagger}$ | 4 | .999 | .999 | +.000 |
| | Thyroid$^{\dagger}$ | 3 | 1.00 | 1.00 | −.000 |
| | Thyroid$^{\dagger}$ | 4 | 1.00 | 1.00 | −.000 |
| | MIMIC-III | 3 | .800 | .798 | −.003 |
| | MIMIC-III | 4 | .800 | .793 | −.007 |
| | eICU | 3 | .786 | .783 | −.004 |
| | eICU | 4 | .786 | .783 | −.003 |
| | MagicTelescope | 3 | .934 | .933 | −.000 |
| | MagicTelescope | 4 | .934 | .932 | −.002 |
| | house_16H | 3 | .951 | .951 | −.001 |
| | house_16H | 4 | .951 | .950 | −.001 |
| | pol | 3 | .997 | .997 | −.000 |
| | pol | 4 | .997 | .997 | −.000 |
| | COMPAS | 3 | .776 | .778 | +.002 |
| | COMPAS | 4 | .776 | .778 | +.002 |
| | RoadSafety | 3 | .830 | .829 | −.001 |
| | RoadSafety | 4 | .830 | .827 | −.003 |
| | Jannis$^{\dagger}$ | 3 | .813 | .811 | −.002 |
| TabDPT | Jannis$^{\dagger}$ | 4 | .813 | .812 | −.001 |
| | Higgs | 3 | .734 | .734 | +.000 |
| | Higgs | 4 | .734 | .733 | −.001 |
| | Covertype$^{\ddagger}$ | 3 | .942 | .941 | −.001 |
| | Covertype$^{\ddagger}$ | 4 | .942 | .940 | −.002 |
| | CTG$^{\dagger}$ | 3 | .996 | .998 | +.002 |
| | CTG$^{\dagger}$ | 4 | .996 | .998 | +.002 |
| | Thyroid$^{\dagger}$ | 3 | .998 | .998 | −.000 |
| | Thyroid$^{\dagger}$ | 4 | .998 | .998 | +.000 |
| | MIMIC-III | 3 | .761 | .759 | −.002 |
| | MIMIC-III | 4 | .761 | .758 | −.003 |
| | eICU | 3 | .757 | .757 | −.000 |
| | eICU | 4 | .757 | .757 | +.000 |
| | MagicTelescope | 3 | .927 | .926 | −.001 |
| | MagicTelescope | 4 | .927 | .921 | −.006 |
| | house_16H | 3 | .954 | .952 | −.003 |
| | house_16H | 4 | .954 | .950 | −.004 |
| | pol | 3 | .998 | .998 | −.001 |
| | pol | 4 | .998 | .998 | −.000 |
| | COMPAS | 3 | .767 | .766 | −.000 |
| | COMPAS | 4 | .767 | .767 | +.001 |
| | RoadSafety | 3 | .805 | .800 | −.005 |
| | RoadSafety | 4 | .805 | .803 | −.001 |
| | Jannis$^{\dagger}$ | 3 | .828 | .823 | −.004 |
| Mitra | Jannis$^{\dagger}$ | 4 | .828 | .821 | −.007 |

| | Dataset | Width | Full AUROC | Skip AUROC | $\triangle$AUROC |
|---|---|---|---|---|---|
| | | | *Table 2 continued* | | |
| Model | Dataset | Width | Full AUROC | Skip AUROC | $\triangle$AUROC |
| | Higgs | 3 | .773 | .771 | $-.002$ |
| | Higgs | 4 | .773 | .769 | $-.004$ |
| | Covertype[‡] | 3 | .927 | .927 | $-.000$ |
| | Covertype[‡] | 4 | .927 | .927 | $-.001$ |
| | CTG[†] | 3 | .998 | .997 | $-.001$ |
| | CTG[†] | 4 | .998 | .998 | $+.000$ |
| | Thyroid[†] | 3 | .999 | .999 | $-.000$ |
| | Thyroid[†] | 4 | .999 | .999 | $-.000$ |
| | MIMIC-III | 3 | .781 | .780 | $-.001$ |
| | MIMIC-III | 4 | .781 | .779 | $-.002$ |
| | eICU | 3 | .772 | .770 | $-.001$ |
| | eICU | 4 | .772 | .768 | $-.004$ |

[†] Multi-class (macro-OVR AUROC; CTG=3-class, Thyroid=3-class, Jannis=4-class). [‡] Covertype (7-class).

## B.3. Skip-position consistency (per dataset)

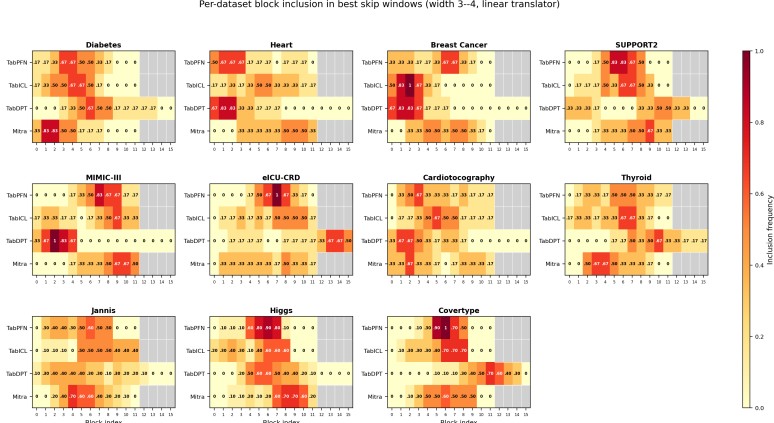

*Figure 5.* Per-dataset block inclusion frequency in best skip windows (width 3–4, linear translator). Gray cells indicate blocks beyond the model's depth (*e.g.*, blocks 12–15 for 12-block models). Cells annotated "0" (lightest yellow) indicate that the block was never selected in the best skip window for that dataset. TabICL and TabDPT show near-uniform inclusion across all datasets, confirming that the aggregate position-insensitivity in Figure 3 is not an averaging artefact.

Per-dataset inclusion frequencies confirm the aggregate pattern (Figure 3): TabICL and TabDPT are near-uniform on each dataset, TabPFNv2 clusters at blocks 7–9 most strongly on MIMIC-III, and Mitra avoids block 0 across all datasets.

## B.4. Extreme compression and naive layer dropping

The Linear (intermediate, "8/12" or "11/16"), Approx. max, and Identity rows of Table 1 provide the per-architecture identity-vs-linear contrast at maximum compression. The retained block is selected by calibration-set AUROC. For Tab-DPT it lands on **block 0** on 11/12 datasets and for TabICL on 11/12, confirming that under single-axis self-attention block 0 carries the dominant computation. Under the per-feature translator (TabPFNv2, Mitra), the retained block varies by dataset (TabPFNv2 1/12, Mitra 3/12 retain block 0): the per-feature parameterisation equalises the contribution of different blocks, so any contiguous window can be replaced.

TabICL retains 1 of 12 blocks within $|\Delta| \leq 3.1\%$ on every dataset evaluated, including multi-class Jannis (4-class) and Covertype (7-class). TabDPT keeps 1 of 16 within $|\Delta| \leq 1.0\%$ on every clinical dataset (worst $-4.2\%$ on Higgs). TabPFNv2 and Mitra, under the per-feature translator appropriate to their dual-axis architecture (Section A.5), retain 1 of 12 blocks within $|\Delta| \leq 1.5\%$ on the multi-class clinical datasets (CTG, Thyroid) and within $-6\%$ to $-9\%$ on the binary ICU mortality datasets (MIMIC, eICU); they trail TabICL/TabDPT by 9–19 AUROC points on the larger TabZilla benchmarks (Jannis, Higgs, Covertype). At moderate compression (4 of 12 retained, "Linear (8/12)"), TabPFNv2 matches or beats its full model on 5 of 12 datasets and is within $0.6\%$ on every clinical dataset; Mitra is within $1.5\%$ on every clinical dataset. Identity dropping at the same compression collapses every model to AUROC $\in [0.27, 0.91]$ depending on architecture and dataset (median around $0.5$), isolating what the translator learns: a low-rank linear correction that maps early representations into the subspace expected by later blocks.

### B.5. NLP comparison: attention entropy

To contextualise tabular attention behaviour, we compare attention entropy profiles between TabICL on a medical (MIMIC-III) and a non-medical (Higgs) benchmark and Qwen3-4B (Yang et al., 2025), a 36-layer language model with grouped-query attention (32 query / 8 KV heads), evaluated on *two* corpora: WikiText-2 and IMDB movie reviews. This 2-vs-2 comparison guards against the contrast being an artefact of any single dataset on either side.

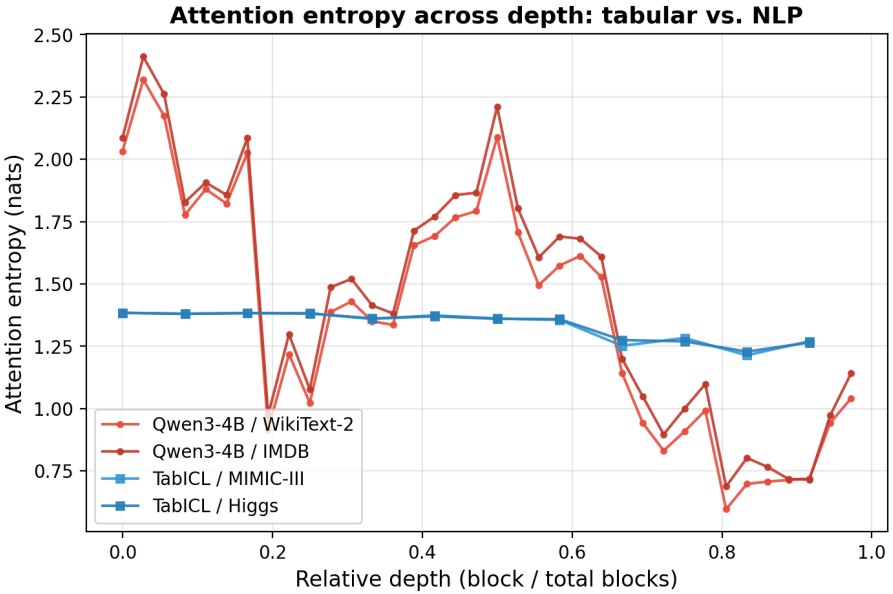

*Figure 6.* Per-layer mean attention entropy across two tabular settings (TabICL on MIMIC-III and Higgs) and two NLP corpora (Qwen3-4B on WikiText-2 and IMDB). Tabular curves are flat across blocks and on both datasets; NLP curves vary non-trivially with depth on both corpora—high in early layers, a dip around layers 7–9, recovery in mid layers, and a marked decline in the last quarter.

TabICL's per-block mean entropy is essentially flat on both MIMIC-III and Higgs (range $\approx 1.23$–$1.38$ nats, std $\leq 0.06$ on each), consistent with persistent, weakly differentiated attention at every depth and across qualitatively different inputs. Qwen3-4B is qualitatively different on both NLP corpora: per-layer entropy varies substantially across the 36 layers (std $\approx 0.4$–$0.5$ nats), with high entropy in the early layers, a sharp dip around layers 7–9, recovery in the mid layers, and a pronounced decline in the last quarter. The contrast is not in the absolute entropy value (which depends on sequence length, $\mathcal{H}_{\max} = \ln S$) but in its *flatness*: tabular blocks all sit at essentially the same entropy regardless of dataset, while NLP blocks span a wide, depth-dependent range that is consistent across corpora, indicative of depth-wise specialisation.

### B.6. Regression results

To verify that block redundancy is not an artefact of classification, we evaluate block approximation on three regression tasks (Table 3): two from the Grinsztajn medium-numerical regression tier (house_sales, predicting King County sale price; superconduct, predicting critical temperature) plus an ICU length-of-stay regression on MIMIC-III. Across all three datasets and all four ICL foundation models with the linear-translator protocol, mild compression (widths 3–4) is essentially lossless: on house_sales every model degrades by $|\Delta R^2| \leq 0.017$ at width 3; on superconduct by $\leq 0.022$; on MIMIC-LOS TabICL is near-lossless at $\Delta R^2 = -0.004$. The finding that block redundancy extends beyond classification across heterogeneous regression domains (real-estate, materials science, hospital outcomes) confirms it is a structural property of TFMs, not an artefact of binary decision boundaries.

## C. Ablation Studies

### C.1. Per-patient redundancy and fairness

A natural clinical concern is whether block approximation disproportionately harms certain patient subpopulations. We compute per-sample $\varepsilon(0,1)$ by fitting a global linear translator $\mathbf{W}^*$ on all test samples and evaluating per-patient residuals.

*Table 3.* Regression block skipping ($R^2$). Best skip start per width selected by calibration-set $R^2$.

| Dataset | Model | Full $R^2$ | Skip ($w{=}3$) | Skip ($w{=}4$) |
|---|---|---|---|---|
| MIMIC-III LOS | TabPFN | .172 | .134 | .110 |
| | TabICL | .200 | **.196** | **.196** |
| | TabDPT | .144 | .131 | .134 |
| | Mitra | .169 | .164 | .165 |
| house_sales | TabPFN | .890 | **.884** | .881 |
| | TabICL | .888 | .883 | .880 |
| | TabDPT | .872 | .855 | .846 |
| | Mitra | .884 | .883 | **.883** |
| superconduct | TabPFN | .883 | .861 | .843 |
| | TabICL | .888 | **.882** | **.879** |
| | TabDPT | .854 | .845 | .825 |
| | Mitra | .835 | .833 | .817 |

*Table 4.* Per-patient $\varepsilon(0,1)$ stratified by outcome. $\Delta = \varepsilon_{\text{survived}} - \varepsilon_{\text{died}}$. Negligible differences confirm that compression does not disproportionately affect any clinical subgroup.

| Model | Dataset | Survived $\varepsilon$ | Died $\varepsilon$ | $\Delta$ |
|---|---|---|---|---|
| TabPFNv2 | MIMIC-III | .241 | .282 | $-.041$ |
| | eICU | .263 | .300 | $-.038$ |
| TabICL | MIMIC-III | .030 | .034 | $-.004$ |
| | eICU | .032 | .034 | $-.003$ |
| TabDPT | MIMIC-III | .079 | .088 | $-.009$ |
| | eICU | .096 | .092 | $+.004$ |
| Mitra | MIMIC-III | .170 | .182 | $-.012$ |
| | eICU | .179 | .181 | $-.002$ |

Across all four models, the difference in per-patient $\varepsilon(0,1)$ between outcome classes is negligible ($|\Delta| \leq 0.041$ on MIMIC-III, $\leq 0.038$ on eICU-CRD). TabPFNv2 shows the largest class difference on both MIMIC-III and eICU ($\Delta \approx -0.04$), meaning deceased patients require slightly more first-block transformation, but the *subsequent* blocks remain redundant for both groups ($\varepsilon(1,2) < 0.05$). Redundancy is universal, not patient-dependent: no subpopulation is disadvantaged by compression.

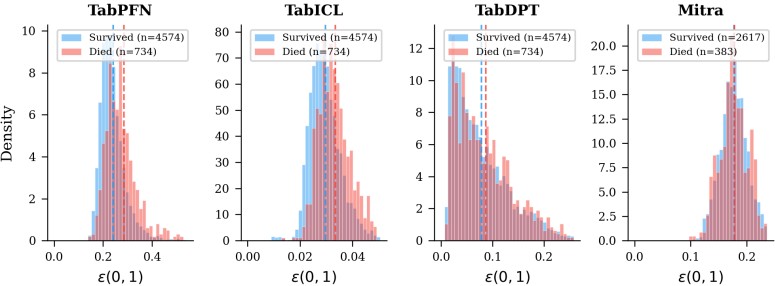

*Figure 7.* Distribution of per-patient $\varepsilon(0,1)$ on MIMIC-III, split by outcome (survived vs. died). Dashed lines show class means. Distributions overlap almost completely, confirming block redundancy is not severity-dependent.

## C.2. Robustness under feature corruption

We test compressed models under random feature corruption (10–50% of features replaced with column-wise random values). Table 5 reports the full sweep on the eICU-CRD ICU mortality benchmark; we additionally re-ran the same protocol at 50% corruption on the five Grinsztajn datasets and confirmed the same pattern: at 50% corruption the maximum $|\Delta\text{AUROC}|$ across all (model, dataset) cells is $0.044$ (TabPFN on pol), and on several cells the affine translator slightly *exceeds* the full model (TabICL on MagicTelescope $\Delta = +0.003$, on pol $\Delta = +0.004$, on road-safety $\Delta = -0.005$; Mitra

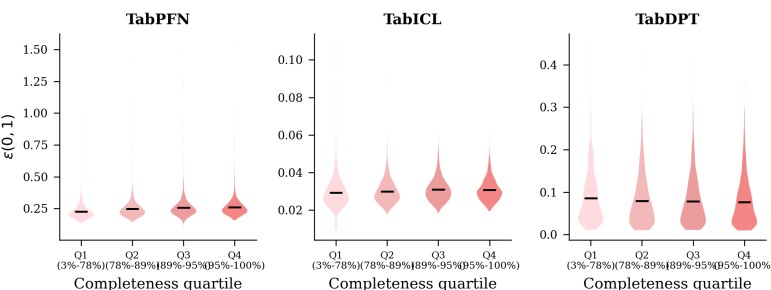

*Figure 8.* Per-patient $\varepsilon(0, 1)$ stratified by feature completeness quartile (Q1 = sparsest, Q4 = densest) on MIMIC-III. No quartile gradient is observed—patients with sparse records do not require more blocks than those with complete records.

on road-safety $\Delta = +0.003$), suggesting the closed-form translator acts as a mild regulariser under input noise.

*Table 5.* Robustness under test-time feature corruption on eICU-CRD (mean over 5 corruption seeds). $\Delta$AUROC = Skip $-$ Full.

| Model | Corruption | Full AUROC | Skip AUROC | $\Delta$AUROC |
|---|---|---|---|---|
| TabPFNv2 | 0% | .758 | .732 | $-.026$ |
| | 10% | .748 | .722 | $-.025$ |
| | 25% | .730 | .707 | $-.023$ |
| | 50% | .691 | .672 | $-.019$ |
| TabICL | 0% | .786 | .782 | $-.004$ |
| | 10% | .772 | .763 | $-.009$ |
| | 25% | .752 | .739 | $-.013$ |
| | 50% | .717 | .699 | $-.018$ |
| TabDPT | 0% | .749 | .751 | $+.002$ |
| | 10% | .736 | .739 | $+.004$ |
| | 25% | .721 | .728 | $+.007$ |
| | 50% | .684 | .693 | $+.008$ |
| Mitra | 0% | .769 | .769 | $-.000$ |
| | 10% | .763 | .765 | $+.002$ |
| | 25% | .749 | .755 | $+.006$ |
| | 50% | .717 | .731 | $+.013$ |

On eICU-CRD, all four models' compressed variants closely track or outperform the full model under increasing corruption. TabDPT's skip model outperforms the full model at every corruption level ($\Delta = +0.008$ at 50%); Mitra's skip model is *more robust* than the full model at 10–50% corruption ($\Delta = +0.013$ at 50%); TabICL stays within $|\Delta| \leq 0.018$ at every level, and TabPFNv2's residual is largest but bounded ($|\Delta| \leq 0.026$ everywhere).

### C.3. Structured clinical missingness

Real clinical missing patterns are structured (whole device fails, lab batch lost) rather than random. We define clinical scenarios that zero out entire feature groups (all labs, all vitals, all variability statistics, all severity scores) and re-evaluate.

On MIMIC-III, the "drop all variability" scenario (81% of features zeroed) is most challenging—TabPFNv2 and TabICL degrade by $\sim 0.05$, while TabDPT and Mitra remain within 0.02. Under realistic single-group scenarios (drop labs or vitals), TabICL/TabDPT/Mitra skip models are within 1.5% of full across both clinical datasets.

### C.4. Early-exit baseline

A natural alternative to block approximation is direct classification from intermediate representations: *early exit* discards the original pre-trained head and trains a new classifier on $\mathbf{H}_l$. We sweep *every* block per model.

We then run a head-to-head comparison: at each depth $l$, early exit fits logistic regression on $\mathbf{H}_l$, while block approximation bridges $\mathbf{H}_l \to \mathbf{H}_L$ with a linear translator and feeds the result through the original pre-trained head—same calibration set, same intermediate representation; only the classifier differs (refitted vs. retained).

*Table 6.* Structured clinical missingness: Full vs. Skip AUROC under realistic missing-data scenarios. $\Delta$ = Skip $-$ Full.

| Model | Scenario | % dropped | Full | Skip | $\Delta$ |
|---|---|---|---|---|---|
| | *MIMIC-III* | | | | |
| TabPFNv2 | Drop all labs | 54% | .644 | .616 | −.028 |
| | Drop all vitals | 43% | .748 | .734 | −.015 |
| | Drop all variability | 81% | .722 | .675 | −.047 |
| TabICL | Drop all labs | 54% | .679 | .679 | −.001 |
| | Drop all vitals | 43% | .779 | .777 | −.002 |
| | Drop all variability | 81% | .756 | .707 | −.048 |
| TabDPT | Drop all labs | 54% | .657 | .657 | −.000 |
| | Drop all vitals | 43% | .736 | .733 | −.003 |
| | Drop all variability | 81% | .706 | .687 | −.019 |
| Mitra | Drop all labs | 54% | .665 | .657 | −.007 |
| | Drop all vitals | 43% | .760 | .760 | −.001 |
| | Drop all variability | 81% | .691 | .728 | +.038 |
| | *eICU-CRD* | | | | |
| TabPFNv2 | Drop all labs | 60% | .667 | .658 | −.009 |
| | Drop all vitals | 38% | .743 | .745 | +.002 |
| | Drop lab variability | 45% | .720 | .724 | +.004 |
| | Drop vital variability | 28% | .748 | .750 | +.001 |
| | Drop all variability | 73% | .701 | .700 | −.001 |
| TabICL | Drop all labs | 60% | .680 | .691 | +.011 |
| | Drop all vitals | 38% | .764 | .764 | +.000 |
| | Drop lab variability | 45% | .746 | .724 | −.021 |
| | Drop vital variability | 28% | .773 | .773 | +.000 |
| | Drop all variability | 73% | .728 | .697 | −.032 |
| TabDPT | Drop all labs | 60% | .632 | .634 | +.002 |
| | Drop all vitals | 38% | .735 | .742 | +.008 |
| | Drop lab variability | 45% | .724 | .731 | +.007 |
| | Drop vital variability | 28% | .732 | .741 | +.009 |
| | Drop all variability | 73% | .682 | .677 | −.005 |
| Mitra | Drop all labs | 60% | .673 | .681 | +.008 |
| | Drop all vitals | 38% | .749 | .750 | +.001 |
| | Drop lab variability | 45% | .725 | .738 | +.013 |
| | Drop vital variability | 28% | .756 | .756 | +.000 |
| | Drop all variability | 73% | .704 | .725 | +.020 |

Block approximation matches or exceeds early exit in the large majority of cells across all four models. The advantage holds even at very early depths ($l = 1, 2$) for TabICL/TabDPT, where skip already approaches full-model AUROC but a fresh classifier on the same $\mathbf{H}_l$ loses 5–10 points (*e.g.* TabICL/MIMIC at $l = 1$: skip .787 vs. probe .733; TabDPT/eICU at $l = 1$: skip .752 vs. probe .675). Retaining the pre-trained head and bridging with a linear translator is uniformly preferable to discarding the head and refitting, except at extreme compression for TabPFNv2/Mitra (the two models with higher residual transformation), where the translator residual eventually dominates and refitting becomes the better option.

Re-running the early-exit probe on the five Grinsztajn datasets (TabPFNv2, TabICL, TabDPT; all probe depths $l = 1 \ldots L$) confirms the pattern outside the clinical regime: across the resulting 147 (model, dataset, depth) cells, the probe is worse than the full model in 142/147 (96.6%). Worst-case probe degradations are $\Delta$AUROC$= -0.31$ (TabPFNv2 at $l = 1$ on MagicTelescope, where the early head has not yet seen meaningful representations) and $\Delta$AUROC$= -0.13$ (TabICL on COMPAS at $l = 7$). The conclusion that early exit is uniformly worse than retaining the pre-trained head holds across both clinical and non-medical Grinsztajn benchmarks.

## D. Related Work

**GBDTs vs. tabular transformers.** Grinsztajn et al. (2022) shows that tree-based models outperform deep learning on typical tabular data, attributing this to lack of rotation invariance and uninformative features. McElfresh et al. (2023) introduced TabZilla and showed neural nets beat GBDTs only on specific subsets. Jayawardhana et al. (2025) recently demonstrated that combining foundation models with decision trees closes this performance gap across sample sizes. Our analysis offers a complementary explanation: tabular transformers waste over 90% of their depth, so their effective capacity is closer to a 1–2 block model than the architecture suggests.

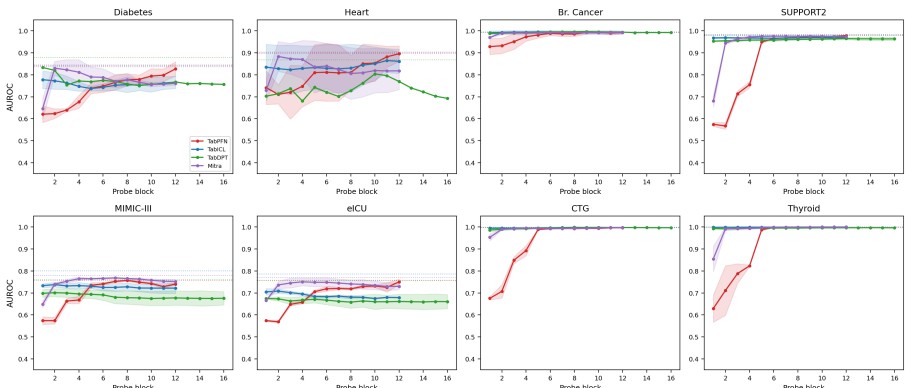

*Figure 9.* Early-exit logistic-regression probe AUROC vs. probe block index. Dotted horizontal lines show the corresponding full-model AUROC. TabPFNv2 (red) rises monotonically with depth; TabICL (blue) and TabDPT (green) are non-monotonic. Mitra (purple) shows an intermediate pattern: probe AUROC rises quickly in the first few blocks then plateaus, with best probes at early depths (L2–L4) on most datasets.

*Table 7.* Head-to-head AUROC at three reference depths: probe (early-exit logistic regression on $\mathbf{H}_l$) vs. skip (linear translator $\mathbf{H}_l \rightarrow \mathbf{H}_L$, original head retained). **Bold** = winner at that cell.

| | | | $l = 1$ | | $l = \lfloor L/2 \rfloor$ | | $l = L$ | |
|---|---|---|---|---|---|---|---|---|
| Model | Dataset | Full | probe | skip | probe | skip | probe | skip |
| TabPFNv2 | MIMIC-III | .758 | **.574** | .547 | .742 | **.743** | .740 | **.758** |
| | eICU | .758 | .574 | **.602** | .719 | **.743** | .750 | **.758** |
| | CTG[†] | .998 | **.675** | .498 | **.992** | .956 | .997 | **.998** |
| | Thyroid[†] | 1.00 | **.629** | .547 | **.997** | .986 | .999 | **1.00** |
| TabICL | MIMIC-III | .800 | .733 | **.787** | .725 | **.798** | .722 | **.800** |
| | eICU | .786 | .705 | **.774** | .683 | **.783** | .679 | **.786** |
| | CTG[†] | .999 | .994 | **.999** | .996 | **.999** | .998 | **.999** |
| | Thyroid[†] | 1.00 | .998 | **.999** | .999 | **.999** | 1.00 | **1.00** |
| TabDPT | MIMIC-III | .761 | .698 | **.751** | .678 | **.758** | .676 | **.761** |
| | eICU | .757 | .675 | **.752** | .658 | **.756** | .661 | **.757** |
| | CTG[†] | .997 | .986 | **.996** | .997 | **.997** | .997 | **.997** |
| | Thyroid[†] | .998 | .993 | .990 | .995 | **.998** | .996 | **.998** |
| Mitra | MIMIC-III | .778 | .645 | **.645** | .763 | **.770** | .753 | **.778** |
| | eICU | .770 | .664 | **.670** | .746 | **.763** | .726 | **.770** |
| | CTG[†] | .998 | **.967** | .801 | .992 | **.998** | .996 | **.998** |
| | Thyroid[†] | .999 | **.853** | .804 | **.999** | .998 | .999 | **.999** |

[†]Three-class datasets (macro-averaged OVR AUROC).

**Tabular transformers and foundation models.** Tabular transformers include FT-Transformer (Gorishniy et al., 2021), SAINT (Somepalli et al., 2021), ExcelFormer (Chen et al., 2023), and TabFlex (Zeng et al., 2025). TabPFNv2 (Hollmann et al., 2025), TabICL (QU et al., 2025), TabDPT (Ma et al., 2026), and Mitra (Zhang et al., 2026) are recent ICL foundation models with diverse pre-training and attention designs. TabNet (Arik et al., 2019) uses sequential sparse attention to select different features at each step; such explicit per-step routing may mitigate block-level redundancy and is an interesting target for future analysis.

**Transformer redundancy.** Dalvi et al. (2020) found 85–92% of BERT/XLNet neurons redundant. (Jacobs et al., 2026) rewrites vision transformers with $k \ll L$ recurrent blocks; Token Merging (Bolya et al., 2023) and SLEB (Song et al., 2024) exploit token and block redundancy. Block-level translators (Cannistraci et al., 2024) fit training-free linear maps between block representations. LoRA (Hu et al., 2022) demonstrated that pre-trained weight updates reside in a low-rank subspace; our finding that a linear translator suffices to bridge skipped blocks is conceptually related, excess depth is captured by a linear approximation.

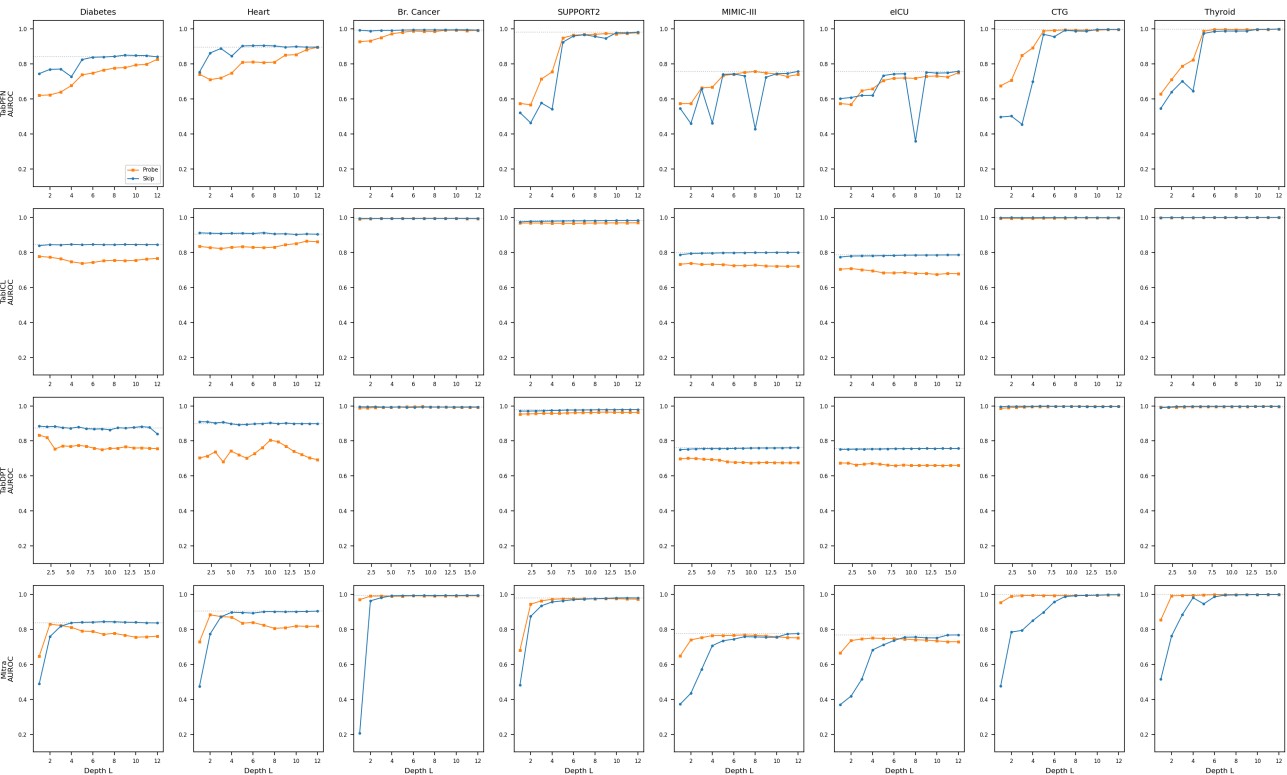

*Figure 10.* Block approximation vs. early-exit at each depth $l$, on the same intermediate representation. Block approximation keeps the pre-trained head and bridges $\mathbf{H}_l \to \mathbf{H}_L$ with a linear translator; early exit replaces the head with a logistic regression fit on $\mathbf{H}_l$. Gray dotted line is the full-model AUROC.

