# OpenReview forum: "Tabular Foundation Models Are Effectively Shallow"
_ICML.cc/2026/Workshop/FMSD — FMSD @ ICML 2026 Poster_

### Official Review · Reviewer_Dn4w · 2026-05-18
**Call in the right direction but calibration requirements can be a blocker**

**Rating:** 6
**Confidence:** 4

**Review:**

The authors present an investigation into redundancy (block-wise) and overparameterization of TFMs.
They evaluate 4 architectures on 15 datasets adopting "training-free" closed form linear translator approach from the TOAST framework.
This allows to bypass intermediate transformer blocks by fitting an OLS linear map between early and late hidden representations.
In evaluations they show that a large fraction of a model depth can be dropped with minimal downstream performance impact for many tasks.

The paper addresses a very timely and pragmatic topic, provocatively questioning the current trend that seems to be to apply the same heavy hammer (TFMs) to every nail: Should we use the same massive and heavy transformer architecture for structured data without checking if the depth is necessary?
Adoption of the TOAST framework is sound and deriving the linear map can be done efficiently. Results on most tasks look promising. Paper is written well. Direct relevance to the workshop is given.

The introduction frames the method in the context of resource-constrained and data-scarce environments. However, the pipeline has some contradictions in smaller data settings: To find the optimal block skipping window, downstream performance (e.g. AUROC) is optimized on a calibration split of e.g. 500 samples. But especially in small data settings, having access to additional 500 labelled samples might be completely infeasible - and solely keeping them aside as a validation split for determining optimal block skipping window seems wasteful.
Looking at the evaluation, it would be interesting to compare to model compression baselines (e.g. structural pruning or distillation).
Moreover, the paper claims to allow for reducing inference latency and memory cost but the paper does not present an analysis of wall-clock times or memory footprints. Additionally, as I understand, TabPFNv2 is evaluated with an ensemble size of 1, but this can result in weaker results. Finally, on some tasks performance drops for Approx Max. seem to be rather severe than "gentle" (e.g. Mitra on MIMIC). I believe, this should be discussed more fairly.

Questions / comments:
* Does the linear translator break column permutation invariance of TFMs?
* Tabular data often suffers from multicolinearity of features. When intermediate representations become correlated, would it make sense to use ridge regularization to guarantee numerical stability when solving the OLS?
* As an alternative to per-dataset selection / calibration, can we select windows on a (set or single) source dataset(s) and generalize zero-shot via meta-learning techniques to new datasets?
* The concrete selection of tasks from each benchmark feels somewhat arbitrary - maybe this presentation can be improved in the future.

The paper discusses a timely property of TFMS, namely structural overparameterization.
Main blockers for me to raise my score is the paradox of requiring a substantial amount of labelled calibration data for window selection which contradicts the scope of optimizing for data-scarce, zero-shot environments.

---

### Official Review · Reviewer_BRr5 · 2026-05-20
**Depth reduction in tabular foundation models using linear translation between layers.**

**Rating:** 6
**Confidence:** 4

**Review:**

### Summary
The authors show that tabular foundation models can be reduced in depth using linear translation between layers, without significant loss of downstream performance across different domains.

### Strengths
The training free least squares approach for linear translation between layers highlights the adaptability of this method to tabular foundation models and therefore showing the potential to reduce the often expensive inference of TFMs.

I appreciate the comparison of the approach on different foundation models highlighting that internal behaviors differ between models and therefore the proposed technique behaves differently for different models.

### Areas for Improvement
Importantly, there is already existing works that explore the contribution of intermediate layers and general inference dynamics in tabular foundation models (https://arxiv.org/abs/2511.15432, https://arxiv.org/pdf/2605.06510) as well as a technique for reducing inference cost (https://arxiv.org/abs/2506.21387) which the authors do not seem to consider in their work, though strongly related.

The evaluation protocol is limited with 15 datasets across diverse benchmarks. A comparison of how the approach scales for different dataset sizes would be very beneficial to better understand the generality of the approach for all types of downstream tasks.

The method is introduced to reduce inference complexity of tabular foundation models, however the actual gains/losses w.r.t. inference complexity are never discussed. How does the proposed method affect inference efficiency, specifically considering that we have to run a separate forward pass for the calibrations set first. Even though the calibration set is much smaller than the full dataset it would be interesting to see the full joined runtime of the method, in comparison to the standard forward pass.

### Detailed Comments
See areas for improvement.  A more detailed comparison to existing works in that direction with specific focus on tabular foundation models would help to better embedd this approach into existing related work. Further a more detailed evaluation, specifically on datasets of various sizes would be very helpful. A comparison of actual runtimes, also given the calibration step, would be important to understand the additional overhead of the proposed method.

### Score
The paper introduces an interesting method for reducing inference complexity in TFMs and therefore fits the topic of the workshop. I will argue for a score of 6 and hope that the comments help the authors improve their work.

---

### Official Review · Reviewer_DCm5 · 2026-05-21
**New empirical insights on the attention mechanisms in TFMs**

**Rating:** 6
**Confidence:** 4

**Review:**

Summary

This paper investigates how modern transformer-based Tabular Foundation Models (TFMs) are overparameterized. The authors demonstrate that most of the performance of TFMs on tabular prediction tasks can be retained while replacing over 90% of the transformer blocks with a linear translator, thereby improving inference latency and reducing memory costs.

Strengths

1.	Empirical insights: The paper identifies fundamental differences between pre-trained LLMs and pre-trained tabular transformers, including differences in attention entropy across layers and the contributions of deeper layers.
2.	Robust performance evaluation: The approach is evaluated across multiple datasets, diverse domains, and various TFMs.

Areas for Improvement

1.	Need for a calibration (validation) set: This approach requires a validation set, which the standard TFM models do not use. This causes a training (and validation) data mismatch between the full TFMs and the compressed models. The full TFMs should be provided with the combined train and validation data as context to ensure an apples-to-apples comparison.
2.	Lack of hardware benchmarks: There is no inference latency or resource usage comparison provided.
3. Does not clearly outperform other fast baselines (GBDTs) - A metric such as average rank over datasets will help to evaluate relative performance.

Detailed Comments

1.	The different compression level labels in Table 1 are confusing. The manuscript would benefit from a more uniform naming scheme.
2.	The details regarding the pair-wise linear approximation error matrix asymmetries and the attention entropy comparison to LLMs are highly interesting. If space permits, consider moving some of these details to the main text.
3.	One of the main themes of the paper is maintaining downstream performance while minimizing compute; therefore, the manuscript would benefit greatly from an inference-time compute (time and resource) comparison.

Justification of Score

This paper demonstrates that over 90% of the blocks in modern TFMs can be replaced with a closed-form linear translator, providing novel empirical evidence and insights regarding the attention mechanisms in TFMs. However, the practical feasibility of this approach for tabular prediction tasks remains to be justified.